# Clinicopathologic and Prognostic Association of GRP94 Expression in Colorectal Cancer with Synchronous and Metachronous Metastases

**DOI:** 10.3390/ijms22137042

**Published:** 2021-06-30

**Authors:** Sumi Yun, Sukmook Lee, Ho-Young Lee, Hyeon Jeong Oh, Yoonjin Kwak, Hye Seung Lee

**Affiliations:** 1Samkwang Medical Laboratories, Department of Diagnostic Pathology, Seoul 06742, Korea; ssum3611@gmail.com; 2Biopharmaceutical Chemistry Major, School of Applied Chemistry, Kookmin University, Seoul 02707, Korea; lees2018@kookmin.ac.kr; 3Department of Nuclear Medicine, Seoul National University Bundang Hospital, Seoul National University College of Medicine, Seongnam 13620, Korea; debobkr@gmail.com; 4Department of Pathology, Seoul National University Bundang Hospital, Seoul National University College of Medicine, Seongnam 13620, Korea; 66109@snubh.org; 5Department of Pathology, Seoul National University Hospital, Seoul National University College of Medicine, Seoul 03080, Korea

**Keywords:** GRP94, tumor-infiltrating lymphocyte, colorectal cancer, prognosis

## Abstract

Patients with advanced colorectal cancer (CRC) with distant metastases have a poor prognosis. We evaluated the clinicopathological relevance of GRP94 expression in these cases. The immunohistochemical expression of GRP94 was studied in 189 CRC patients with synchronous (SM; *n* = 123) and metachronous metastases (MM; *n* = 66), using tissue microarray; the association between GRP94 expression, outcome, and tumor-infiltrating lymphocytes (TILs) was also evaluated. GRP94 was expressed in 64.6% (122/189) patients with CRC; GRP94 positivity was found in 67.5% and 59.1% patients with SM and MM, respectively. In the SM group, high GRP94 expression was more common in patients with a higher density of CD4+ TILs (*p* = 0.002), unlike in the MM group. Survival analysis showed that patients with GRP94 positivity had significantly favorable survival (*p* = 0.030); after multivariate analysis, GRP94 only served as an independent prognostic factor (*p* = 0.034; hazard ratio, 0.581; 95% confidence interval, 0.351–0.961) in the SM group. GRP94 expression was detected in 49.4% of metastatic sites and showed significant heterogeneity between primary and metastatic lesions (*p* = 0.012). GRP94 is widely expressed in CRC with distant metastases; its expression was associated with favorable prognosis in the SM group, unlike in the MM group.

## 1. Introduction

Although the incidence and mortality rates of colorectal cancer (CRC) have improved with early screening and multidisciplinary therapeutic approaches, some patients continue to present with advanced disease; the mortality of these patients remains relatively high [1,2]. There is therefore a clear need to identify markers for better prediction of patient outcomes and treatment responses in advanced CRC.

Glucose-regulated protein 94 (GRP94), a member of the heat shock protein family, is a specialized chaperone protein [3,4], and has been implicated in carcinogenesis [3]. Numerous studies have identified GRP94 to be a chaperone with various essential proteins; it forms complexes with the Wnt signaling cofactor, glycoprotein A repetitions predominant, and several integrins, leading to cell proliferation, epithelial–mesenchymal transition, and tumor progression. Previous studies have reported that GRP94 expression is associated with tumor progression and unfavorable prognosis in lung, breast, and esophageal cancers [5,6,7,8]. In addition, a potential cancer-suppressing function of GRP94 has recently been identified. Surface expression of GRP94 contributes to tumor regression via the modulation of immune surveillance, leading to dendritic cell maturation and differentiation of T lymphocytes [4,9,10]. Despite these findings, the clinicopathologic significance and survival effect of GRP94 remain controversial in patients with CRC [11,12,13,14]. Reports suggest that GPR94 correlates with poorly differentiated histology and advanced Duke stage [11]. Several recent reports have described an association between GRP94 and a favorable prognosis in colon cancer [13,14]. Furthermore, some studies suggest that it may be related to chemoresponse in some types of cancers, and its inhibition could significantly reduce the proliferation of cetuximab-resistant colon cancer cells [15,16,17,18,19].

Based on these findings, we analyzed the expression of GRP94 and evaluated its prognostic significance and association with tumor-infiltrating lymphocytes (TILs) in advanced CRC with distant metastasis.

## 2. Results

### 2.1. Clinicopathologic Features of Colorectal Cancer

The baseline characteristics of all patients with metastatic CRC are described in Appendix A. Briefly, 102 (54.0%) male and 87 (46.0%) female CRC patients with a median age of 60 years (range: 28–93 years) were identified. A total of 141 (74.6%) tumors were located on the left side of the colon, and low-grade differentiation was documented in 164 (86.8%) cases. The most common site of distant metastasis was the liver (*n* = 88), followed by the lung (*n* = 41), peritoneum (*n* = 37), ovary (*n* = 18), and non-regional lymph nodes (*n* = 5).

### 2.2. GRP94 Expression in CRC

We evaluated the immunohistochemical expression of GRP94 in primary CRC tissues (Figure 1); GRP94 expression was detected in 122 cases (64.6%). Overall, 83 (67.5%) and 39 (59.1%) patients demonstrated GPR94 positivity in the SM and MM groups, respectively. *KRAS*, *PIK3CA*, and *BRAF* mutations were found in 101 (53.4%), 26 (13.8%), and 7 (3.7%) cases, respectively; 9 (9/185, 4.9%) cases harbored *HER2* amplifications. CRC patients with GRP94 expression commonly harbor *PIK3CA* mutations (*p* = 0.021). Regarding various clinicopathologic parameters, there were no other significant differences in the entire cohort. In the SM group, GPR92 expression was associated with *PIK3CA* mutations (*p* = 0.026). In addition, it showed a tendency for association with low-grade histology; however, this tendency was not statistically significant (*p* = 0.087). There were no statistically significant differences between GRP94 expression and other clinicopathological factors in either the SM or MM groups (Table 1, Appendix A).

### 2.3. GRP94 Expression Status in Primary CRC, Its Invasive Border, and Metastasis

To investigate the heterogeneity of GRP94 expression between the primary and metastatic sites, we also performed immunohistochemical staining in tissue from the invasive border of the primary tumor and matching metastatic tumor specimens (Appendix A). Cases with tissue loss and insufficient tumor material were excluded from the analysis. Among the 175 samples with invasive borders analyzed, GRP94 expression was found in 108 cases (61.7%); there was no significant discordance in GRP94 expression status. Among 170 specimens from metastatic tissue, 84 (49.4%) revealed GRP94 positivity; this frequency was significantly lower than that observed in primary CRC lesions. Lung metastases and peritoneal seeding demonstrated high GRP94 expression (66.7% [24/36] and 63.6% [21/33], respectively), followed by metastases in the ovary (55.6% [10/18]), liver (35.9% [28/78]), and distant nodes (20.0% [1/5]). Similar trends were observed in the SM group; GRP94 expression was more frequently observed in lung and peritoneal seeding (66.7% and 63.0%, respectively), and all 4 cases of non-regional lymph node metastases tested negative for GRP94. In the MM group, GRP94 expression was found in 66.7% of patients with both, lung metastases and peritoneal seeding (Appendix A). As shown in Appendix A, GRP94 expression was inconsistent between the primary and metastatic sites. In the SM group, GRP94 expression was observed in 59.8% (70/117) and 47.2% (51/108) cases with invasive borders and metastatic sites, respectively. In the MM group, GPR94 positivity was detected in 65.5% (38/58) and 53.2% (33/62) cases with invasive borders and metastatic sites, respectively. Similar to the results in the total cohort, no significant difference was found between the primary tumor and its invasive border; GRP94 was significantly less commonly expressed in the metastatic sites and showed significantly different expression rates compared with those in primary tumors in the SM group. This trend was not observed in the MM group (Appendix A).

With regard to metastatic sites, lung metastases and peritoneal seeding showed high GRP94 expression (66.7% [24/36] and 63.6% [21/33], respectively), followed by the ovary (55.6% [10/18]), liver (35.9% [28/78]), and distant nodes (20.0% [1/5]). Similar trends were observed in the SM group; GRP94 expression was more frequently observed in lung metastases and peritoneal seeding (66.7% and 63.0%, respectively); all 4 cases with non-regional lymph node metastases tested negative for GRP94. In the MM group, GRP94 expression in both, lung metastases and peritoneal seeding was found in 66.7% of patients (Appendix A).

### 2.4. Association between GRP94 Expression and TILs

We explored the association between GRP94 expression and TIL status (Figure 2). The median densities of each immune cell were as follows: CD3, 297.68 (range: 11.64–1658.51); CD4, 97.09 (range: 1.61–969.03); CD8, 108.05 (range: 5.57–1322.19); and Foxp3, 12.71 (range: 0–265.54). In the entire cohort, GRP94 positivity correlated significantly positively with higher infiltration of CD4+ TILs (*p* = 0.011). Furthermore, analysis of the SM group demonstrated the association between GRP94 expression and CD4+ TILs (*p* = 0.002). However, there was no significant difference in TIL density depending on GRP94 expression in either the SM or MM groups.

### 2.5. Survival Analysis

The average survival time of patients was 42.9 months (range, 0.8–104.6 months). In all 189 patients, GRP94 expression did not influence overall survival (OS *p* = 0.384). In the SM group, patients with GRP-positive tumors had significantly improved OS compared with those having GRP94 negative tumors (*p* = 0.030). On multivariate Cox regression analysis, GRP94 expression remained an independent prognostic factor for prolonged OS only in the SM group (hazard ratio [HR], 0.581; 95% confidence interval [CI]: 0.351–0.961, *p* = 0.034). In contrast, no association was found between GRP94 expression and OS in the MM group (*p* = 0.702; Figure 3, Table 2, Appendix A).

### 2.6. Correlation between GRP94 Expression and Genetic Alterations

A recent study by Jeoung et al. found that surface GRP94 inhibitors may contribute to enhanced survival in cetuximab-resistant CRC [19]. As the response to cetuximab may be influenced by the presence of several genetic alterations including *KRAS*, *PIK3CA*, and *BRAF* mutations, and *HER2* amplification, we sought to investigate the association with survival in patients harboring genetic alterations in these genes. In the SM group, 80 of 123 (65.0%) tumors carried one or more genetic alterations in these genes; GRP94 positivity was found in 56 of 80 (70.0%) cases. Although the number of cases was small, GRP94 positive patients also displayed significantly better OS (*p* = 0.009) among those with genetic alterations. Multivariate survival analyses also indicated that the prognostic association of GRP94 positivity was obvious in this subgroup (HR = 0.438, 95% CI: 0.234–0.821, *p* = 0.010). Among the 65 patients in the MM group, one or more genetic alterations were detected in 39 (60.0%) cases. GRP94 positivity was detected in 23 (59.0%) of these patients. Unlike the previous results in the SM group, GRP94 expression was not significantly associated with OS (*p* = 0.766) in CRC with MM and genetic alterations. The detailed results are presented in Figure 4 and Table 3.

## 3. Discussion

In this study, we demonstrated that GRP94 is widely expressed in advanced CRC with synchronous and metachronous metastasis. Although we failed to find an association between GRP94 expression and various clinicopathological parameters, we demonstrated that GRP94 expression correlated significantly with a favorable prognosis on univariate and multivariate analyses. We also observed that GRP94 expression correlated with higher infiltration of CD4+ TILs. Findings for GRP94 positive patients with CRC having SM were similar, with favorable survival; this was not observed in the MM subgroup.

Recent studies have focused on surface-located GRP94 as a therapeutic target. Monoclonal antibodies for the inhibition of GRP94 have been shown to be related to anti-tumoral effects in melanoma and breast cancer [16,20,21]. In addition, in a study using a cetuximab-resistant colon cancer cell line model, antibody-based targeting of surface GRP94 demonstrated therapeutic impact [19]. We investigated the prognostic impact of GRP94 in metastatic CRC harboring genetic alterations associated with chemoresistance to cetuximab. Our results show that GRP94 is still widely expressed in these cancer tissues (66.4%). In particular, its expression was slightly more frequent in the SM group. In this study, the expression status of GRP94 in primary cancers frequently differed from that of the paired distant metastases, especially in the SM group. Therefore, both primary and metastatic cancers should be tested for GRP94 expression if the SM group is tested. In addition, GRP94 expression rates were high in CRC cases with peritoneal seeding or lung metastasis; this makes complete metastasectomy relatively difficult to perform. This may be one of the possible advantages of using GRP94 inhibitors in patients with stage IV CRC.

The link between endoplasmic reticulum stress (ERS) and cancer cell death is intriguing. The endoplasmic reticulum has many biological functions including protein synthesis, transportation, and folding. Many unfolded proteins accumulate beyond the capacity of the ER from a variety of causes, such as inflammation and oxidative stress, that trigger ERS [22,23]. In cancer, it has been suggested that prolonged ERS promotes autophagy and apoptosis, and reduces resistance to chemotherapy via various mechanisms such as downregulation of the PI3K/Akt/mTOR pathway and activation of caspase-dependent apoptosis [22,24,25,26,27]. Fu et al. showed that GRP94 expression, one of the hallmarks of the ERS response, was associated with cancer cell death and reduced chemoresponse in an in vitro model [28]. In this study, we found that GRP94 expression was detected more frequently in *PIK3CA* mutant CRC; further studies are needed to explore the detailed mechanism and function of GRP94 in such regulatory networks in CRC.

As a major chaperone protein, GRP94 is known to participate in the regulation of various biological processes, such as carcinogenesis and cancer progression. It has also been suggested to play a key role in anti-tumor immune responses via immune cell recruitment [4,20]. We analyzed the association between high GRP94 expression and TIL levels and found that the expression of the former was significantly associated with higher infiltration of CD4+ TILs. This association was observed more clearly in the SM than in the MM group. Several studies have suggested that CD4+ T cells are needed to activate CD8+ T cells; their population has also been suggested to have a direct anti-tumor effect. In addition, CD4+ T cells are required for the induction of B cell-mediated cellular immune responses against tumor cells [29,30,31]. Takagi et al. observed that GRP94 was associated with the infiltration of CD8+ and CD4+ T cells via the maturation of dendritic cells; they suggested that this could improve patient survival in cholangiocarcinoma [32]. Baker-LePain et al. also found that GRP94 may influence the activation of CD4+ TILs in a chaperone-independent manner [33]. In our study, patients with higher CD4+ TILs tended to show prolonged survival (data not shown); however, the prolongation was not statistically significant. Interestingly, some previous studies revealed the interactive role of GRP94 with macrophages, another component of the tumor microenvironment. Tumor-derived GRP94 also might induce the maturation of dendritic cells and macrophages, and enhance the macrophage mediate host defense [20,32]. In view of these findings, GRP94 expression may improve prognosis through various mechanisms including recruitment of TILs in the late stage of metastatic CRC, rather than promote the progression of cancer. Although our analysis included a limited number of cases, evaluation of TILs among diverse tumor microenvironment components, and molecular alterations that affect the response to cetuximab, the present study may provide several clues for understanding the role of GRP94 expression in CRC pathogenesis. However, further research studies investigating these mechanisms and prospective studies involving a large cohort of patients with CRC are warranted.

## 4. Materials and Methods

### 4.1. Patient, Tissue Samples, and Tissue Microarray Construction

This study included 189 patients with advanced CRC with distant metastasis, between 2003 and 2009. Patients were divided into two groups, namely, synchronous (SM) and metachronous (MM) metastasis. SM was defined as distant metastasis identified within six months after the diagnosis of primary CRC; MM included patients with distant metastasis identified six months after the initial diagnosis of CRC. Among 189 patients, 123 (65.1%) had SM and 66 (34.9%) had MM. The patients’ clinicopathologic data including those of clinical follow-up, were retrieved from the medical records. All patients underwent surgical resection of the primary CRC without neoadjuvant treatment. For the construction of the tissue microarray (TMA), Hematoxylin and Eosin-stained slides were evaluated by an expert gastrointestinal pathologist. The most representative core measuring 2 mm in diameter was obtained from the primary tumor and transferred to a recipient block. Additional tissue cores were taken from the invasive border of the primary tumor and the paired metastatic lesion [34].

### 4.2. Immunohistochemistry and Interpretation

Immunohistochemical staining was performed on the four micrometer TMA sections. Immunostaining for CD3 (DAKO, Glostrup, Denmark), CD8 (Neomarkers, Fremont, CA, USA), and Foxp3 (Abcam, Cambridge, UK) were performed using a Leica BOND-MAX autostainer. For GRP94 (Abcam, Cambridge, UK) and CD4 (Ventana, Tucson, AZ, USA), immunohistochemical staining was performed using the BenchMark XT platform. For detection of EGFR mutations, we performed immunohistochemistry using the EGFR pharmDx kit (Dako, Glostrup, Denmark) following the manufacturer’s instructions. Moderate to strong immunostaining for GRP94 in more than 25% of tumor cells was defined as GRP94 positivity; EGFR-positivity was defined by the presence of any membranous immunostaining of tumor cells [34]. For quantitative TIL analysis, immunostained TMA slides were scanned using an Aperio ScanScope CS slide scanner (Aperio Technologies, Vista, CA, USA) at 40× magnification. After digitalization, the automated digital image analysis algorithm (ImageScope TM, Aperio Technologies, Vista, CA, USA) of Aperio was used to quantify CD3+, CD4+, CD8+, and Foxp3+ TILs, and the density of each TIL was recorded [35]. The density of each immunoreactive TIL was dichotomized into high and low groups based on the median values.

### 4.3. Microsatellite Instability

The real-time polymerase chain reaction-based test was performed on formalin-fixed paraffin-embedded tissue for the determination of microsatellite instability (MSI status). Using a deoxyribonucleic acid (DNA) automatic sequencer (ABI 3731 Applied Biosystems, Inc., Foster City, CA, USA), the two mononucleotides (BAT-25 and BAT-26) and three dinucleotide markers (D2S123, D5S346, and D17S250) recommended by the National Cancer Institute were evaluated. MSI-high and MSI-low tumors included those with instability in two or more and one markers, respectively. Tumor samples with no instability were classified as microsatellite stable [35].

### 4.4. KRAS, PIK3CA, and BRAF Mutational Analysis

Mutations in *the*
*KRAS*, *PIK3CA*, and *BRAF* genes were detected using a real-time polymerase chain reaction. Briefly, the tumor area containing more than 60% of the tumor cells was microdissected from 8-μm thick formalin-fixed paraffin-embedded sections. After DNA extraction using the Cobas DNA Sample Preparation kit (Roche, Basel, Switzerland), mutation analysis was performed for each gene using the Cobas 4800 Mutation test (Roche Molecular Systems, Inc., Branchburg, NJ, USA), which detected *KRAS* mutations in codons 11, 13, and 61, *PIK3CA* mutation in exons 1, 4, 7, 9, and 20, and the *BRAF* V600E mutation [34].

### 4.5. HER2 Amplification

*HER2* amplification was determined using silver in situ hybridization (SISH) on a Ventana XT automated SISH instrument; INFORM HER2 DNA and INFORM Chromosome 17 (CEP17) probes (Ventana Medical Systems, Tuscon, AZ, USA) were used as previously described [34]. SISH was performed in 185 cases for *HER2* amplification; cases with *HER2/CEP17* ratios ≥2.0, were classified as HER2 positive, following the American Society of Clinical Oncology/College of American Pathologists guidelines for HER2 testing in gastric cancer [34,36].

### 4.6. Statistical Analysis

Statistical comparisons between groups were performed using the chi-square, Fisher’s exact, and McNemar’s tests for categorical variables, and the Mann–Whitney U test for continuous variables. The Kaplan–Meier method with the log-rank test was used for survival analysis. the Significant prognostic factors for patient survival were determined using univariate and multivariate Cox proportional hazard models. All data analyses were performed using the Statistical Package for the Social Sciences software package (version 21.0; IBM Corp., Armonk, NY, USA); a *p*-value of <0.05 was regarded as statistically significant.

## 5. Conclusions

In this cohort, we found that GRP94 expression was commonly detected in advanced CRC with SM and MM; however, significant heterogeneity in GRP94 expression was observed between the primary tumor and metastatic site. Additionally, GRP94 positivity correlated with higher infiltration of CD4+ TILs. Its association with patient outcomes suggests that GRP94 expression may contribute to a better prognosis in CRC with SM. Further prospective studies in larger cohorts are needed to validate our findings.

## Figures and Tables

**Figure 1 ijms-22-07042-f001:**
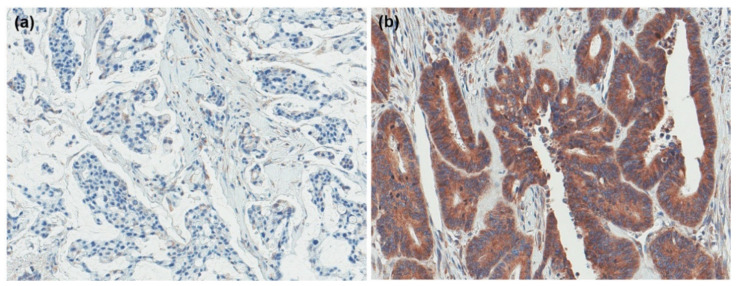
GRP94 expression in colorectal cancer. (**a**) negative (**b**) positive.

**Figure 2 ijms-22-07042-f002:**
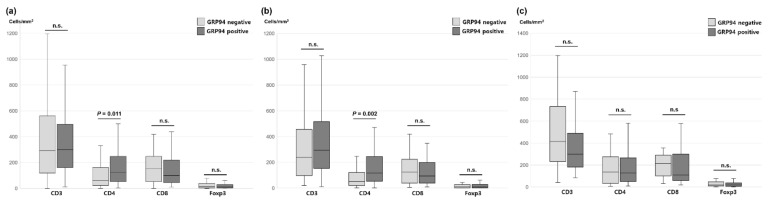
Association between GRP94 expression and TIL density in the (**a**) total, (**b**) SM group, and (**c**) MM group. n.s.: not significant.

**Figure 3 ijms-22-07042-f003:**
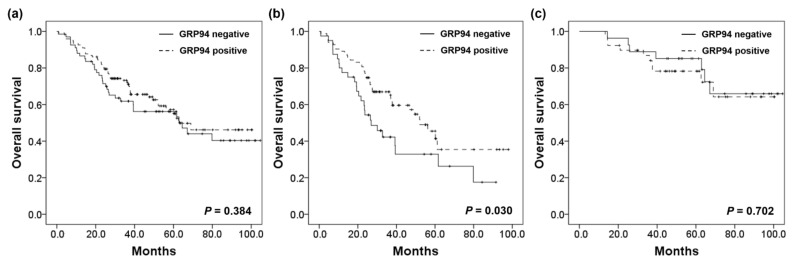
Kaplan–Meier curves of overall survival for advanced colorectal cancer with synchronous metastasis in the (**a**) total cohort, (**b**) SM group, and (**c**) MM group.

**Figure 4 ijms-22-07042-f004:**
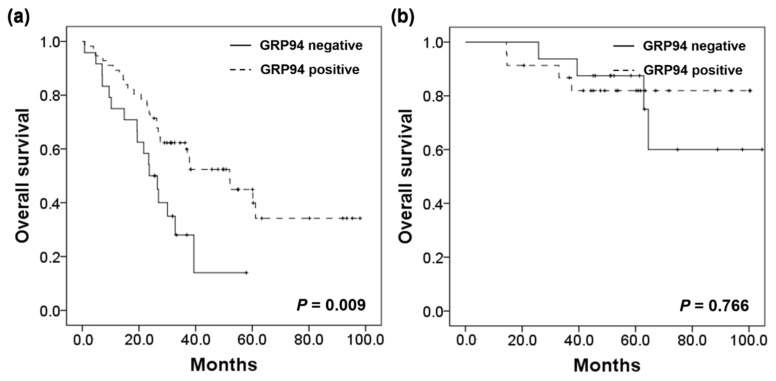
Survival analysis in the (**a**) SM group, and (**b**) MM group harboring *KRAS*, *PIK3CA*, *BRAF* mutations or *HER2* amplification.

**Table 1 ijms-22-07042-t001:** Clinicopathologic characteristics of advanced colorectal cancer with synchronous or metachronous metastasis by GRP94 expression status.

Characteristics	CRC with SM		*p*	CRC with MM		*p*
	GRP94 (−)	GRP94 (+)		GRP94 (−)	GRP94 (+)	
Age			0.341			0.347
<65	29 (35.4%)	53 (64.6%)		17 (45.9%)	20 (54.1%)	
≥65	11 (26.8%)	30 (73.2%)		10 (34.5%)	19 (65.5%)	
Sex			0.399			0.756
M	17 (28.8%)	42 (71.2%)		17 (39.5%)	26 (60.5%)	
F	23 (35.9%)	41 (64.1%)		10 (43.5%)	13 (56.5%)	
Location			0.470			0.748
Right sided	10 (27.8%)	26 (72.2%)		4 (33.3%)	8 (66.7%)	
Left sided	30 (34.5%)	57 (65.5%)		23 (42.6%)	31 (57.4%)	
Differentiation			0.087			1.000
Low grade	31 (29.5%)	74 (70.5%)		24 (40.7%)	35 (59.3%)	
High grade	9 (50.0%)	9 (50.0%)		3 (42.9%)	4 (57.1%)	
Lymphatic invasion			0.394			0.982
Absent	10 (27.0%)	27 (73.0%)		11 (40.7%)	16 (59.3%)	
Present	30 (34.9%)	56 (65.1%)		16 (41.0%)	23 (59.0%)	
Venous invasion			0.416			0.935
Absent	24 (30.0%)	56 (70.0%)		21 (41.2%)	30 (58.8%)	
Present	16 (37.2%)	27 (62.8%)		6 (40.0%)	9 (60.0%)	
Perineural invasion			0.732			0.566
Absent	17 (30.9%)	38 (69.1%)		14 (37.8%)	23 (62.2%)	
Present	23 (33.8%)	45 (66.2%)		13 (44.8%)	16 (55.2%)	
pT stage			0.950			0.748
pT2-3	20 (32.3%)	42 (67.7%)		23 (42.6%)	31 (57.4%)	
pT4	20 (32.8%)	41 (67.2%)		4 (33.3%)	8 (66.7%)	
pN stage			0.217			0.539
pN0	2 (16.7%)	10 (83.3%)		11 (45.8%)	13 (54.2%)	
pN+	38 (34.2%)	73 (65.8%)		16 (38.1%)	26 (61.9%)	
MSI			1.000			0.509
MSS/MSI-low	40 (32.8%)	82 (67.2%)		27 (42.2%)	37 (57.8%)	
MSI-high	0 (0%)	1 (100%)		0 (0%)	2 (100%)	
EGFR			0.497			0.085
Negative	23 (30.3%)	53 (69.7%)		13 (32.5%)	27 (67.5%)	
Positive	17 (36.2%)	30 (63.8%)		14 (53.8%)	12 (46.2%)	
*KRAS* mutation			0.661			0.522
Wild type	20 (34.5%)	38 (65.5%)		11 (36.7%)	19 (63.3%)	
Mutant	20 (30.8%)	45 (69.2%)		16 (44.4%)	20 (55.6%)	
*PIK3CA* mutation			0.026			0.691
Wild type	38 (36.5%)	66 (63.5%)		25 (42.4%)	34 (57.6%)	
Mutant	2 (10.5%)	17 (89.5%)		2 (28.6%)	5 (71.4%)	
*BRAF* mutation			1.000			1.000
Wild type	38 (32.5%)	79 (67.5%		27 (41.5%)	38 (58.5%)	
Mutant	2 (33.3%)	4 (66.7%)		0 (0%)	1 (100%)	
*HER2* amplification			0.268			NA
Negative	38 (34.2%)	73 (65.8%)		26 (40.0%)	39 (60.0%)	
Positive	1 (11.1%)	8 (88.9%)		0 (0%)	0 (0%)	
Any mutation			0.416			0.836
Absent	16 (37.2%)	27 (62.8%)		10 (38.5%)	16 (61.5%)	
Present	24 (30.0%)	56 (70.0%)		16 (41.0%)	23 (59.0%)	

**Table 2 ijms-22-07042-t002:** Multivariate analysis for overall survival.

Factor	Hazard Ratio (95% CI)	Significance
Total		
Lymphatic invasion (present vs. absent)	1.803 (1.001–3.251)	0.050
Neural invasion (present vs. absent)	1.687 (1.039–2.740)	0.035
pT stage (pT4 vs. pT2-3)	1.579 (1.007–2.476)	0.047
pN stage (pN+ vs. pN0)	2.689 (1.231–5.960)	0.015
Synchronous metastasis group		
Age (≥65 vs. <65)	1.978 (1.196–3.271)	0.008
Lymphatic invasion (present vs. absent)	2.014 (1.070–3.791)	0.030
GRP94 (positive vs. negative)	0.581 (0.351–0.961)	0.034
Metachronous metastasis group		
Neural invasion (present vs. absent)	8.082 (2.307–28.319)	0.001

**Table 3 ijms-22-07042-t003:** Multivariate regression model of prognostic factors in synchronous metastasis group harboring *KRAS*, *PIK3CA*, *BRAF* mutations or *HER2* amplification.

Factor	Hazard Ratio (95% CI)	Significance
Synchronous metastasis group		
Age (≥65 vs. <65)	2.449 (1.354–4.431)	0.003
GRP94 (positive vs. negative)	0.438 (0.234–0.821)	0.010

## Data Availability

Data are contained within the article and Appendix A.

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
