# Peer review of "Clinicopathologic and Prognostic Association of GRP94 Expression in Colorectal Cancer with Synchronous and Metachronous Metastases"

_ijms, 2021, doi:10.3390/ijms22137042_

Round 1

Reviewer 1 Report

    The manuscript sent by the authors for publication is very interesting for oncology studies and for understanding the role of the different forms with which metastases occur, elements that suppress the survival of the individual. On the basis of the results obtained by evaluating the immunohistochemical expression of GRP94 it will be possible to construct targeted therapies. I suggest the authors to introduce a table with photographs showing metastasis heterogeneity and immunohistochemical labeling for GRP94.

Reviewer 2 Report

Review for IJMS

Title: Clinicopathologic and prognostic association of GRP94 expression in colorectal cancer with synchronous and metachronous metastases

Authors: Sumi Yun, Sukmook Lee, Ho-Young Lee, Hyeon Jeong Oh, Yoonjin Kwak, Hye Seung Lee

COMMENTS:

The authors have described the correlations they revealed between the GRP94 expression and outcomes of colorectal cancer (CC) with synchronous metastases (SM) or metachronous metastases (MM). According to their observations, the GRP94 expression seems to be associated with more favorable prognosis in the SM group but not in the MM group. As the authors suggested, the found diversity may be due to a higher density of CD4+ tumor-infiltrating lymphocytes in the samples from SM group exhibiting the high expression of GRP94. These findings are interesting and may have a prognostic significance. However, the authors should strengthen their evidence basis and explanations/discussion.  

1) In particular, if the authors suggested the lymphocyte infiltration as a causal factor, the question  arises: have they seen the in situ signs of immunogenic death of CC cells in the samples from SM group? (If not, how it may be explained?)

2) If the authors discuss that endoplasmic reticulum (ER) stress-associated apoptosis might play a role in the observed phenomena, such known pro-apoptotic responses to ER stress as the induction of CHOP and activation of caspases in the samples of CC cells had to be demonstrated in situ or at least discussed.

3) The revealed high expression GRP94 might ensure the calreticulin expression on the CC cell surface followed by phagocytosis of such cancer cells by macrophages (and this may lead to the improved outcome). The authors should discuss (if not examine in situ) such a mechanism in the context of their findings. 
